# Dynamic Mechanical Properties of TC11 Titanium Alloys Fabricated by Wire Arc Additive Manufacturing

**DOI:** 10.3390/ma15113917

**Published:** 2022-05-31

**Authors:** Ze Tian, Haijun Wu, Chengwen Tan, Heng Dong, Meng Li, Fenglei Huang

**Affiliations:** 1State Key Laboratory of Explosion Science and Technology, Beijing Institute of Technology, Beijing 100081, China; tzyhy86@126.com (Z.T.); dongheng789@gmail.com (H.D.); 3120200124@bit.edu.cn (M.L.); huangfl@bit.edu.cn (F.H.); 2School of Materials Science, Beijing Institute of Technology, Beijing 100081, China; tanchengwen@bit.edu.cn

**Keywords:** wire arc additive manufacturing technology, TC11 titanium alloy, strain rate effect, fracture morphology, constitutive model

## Abstract

To study the compressive mechanical properties and failure modes of TC11 titanium alloy fabricated by wire arc additive manufacturing (WAAM) technology in a large strain rate range at room temperature, the quasi-static and dynamic compression tests were carried out. In addition, optical microscopy (OM) and scanning electron microscopy (SEM) were employed to observe the metallographic structure and fracture morphology, respectively. The stress–strain curves in the range of 0.001 s^−1^–4000 s^−1^, original and post-deformation microstructures, macroscopic damage patterns, and microscopic fracture morphology were obtained at two different loading directions, including the scanning and deposition directions, respectively. In uniaxial compression experiments, the material showed little difference in mechanical properties between the scanning and deposition directions, exhibiting a strain rate strengthening effect. However, the strain rate sensitivity of the material under quasi-static loading conditions is much less than that under dynamic loading conditions. In addition, combining the stress–strain curve with the fracture morphology analysis, the plasticity in the scanning direction is better than in the deposition direction. Based on the experimental results, a modified Johnson–Cook (JC) constitutive model considering strain rate sensitivity and the effect of strain rate on strain hardening was proposed, and the parameters were determined using a Multiple Population Genetic Algorithm (MPGA). The obtained constitutive model is in good agreement with the experimental data, which can provide a reference for the engineering numerical calculation of TC11 titanium alloy for WAAM. This study also provides a fundamental databank for the application and design of WAAM TC11 alloy in the manufacturing of large and complex structural parts.

## 1. Introduction

Titanium alloys with high specific strength, good ductility, corrosion resistance, good fracture toughness, and excellent biocompatibility have become one of the most promising metal structural materials in the aviation, aerospace, marine, and medical industries [1,2,3,4,5,6,7,8]. Ti-6.5Al-3.5Mo-1.5Zr-0.3Si is an α + β two-phase titanium alloy (known as TC11 titanium alloy in China and BT9 titanium alloy in Russia) with excellent comprehensive mechanical properties at room and high temperatures [9,10]. TC11 has been extensively used in key structural components of aircraft engines such as compressor disks, blades, and drums [11]. Given the high melting point and strength of titanium alloys, the use of traditional forging, casting, welding, and machining processes for complex-shaped titanium alloy structural parts has the disadvantages of low material utilization rate, variable material mechanical properties, high cost, long processing cycle, difficult processing, and insufficient performance [4,12]. Therefore, there is an urgent need to explore new manufacturing processes for titanium alloys to meet the growing demand for complex-shaped titanium alloy components in several industries. Wire arc additive manufacturing (WAAM) technology uses wire as the raw material. Compared with the technology that uses powder as the raw material, WAAM has lower production costs, higher deposition efficiency, and a higher material utilization rate. The use of arc as a heat source also makes it more cost-efficient than using a laser or electron beam as the heat source additive manufacturing technology. In addition, due to the simple manufacturing environment and unlimited processing space, it can be used for the preparation of large-size molding parts [13,14,15]. With a continued need to improve material efficiency and lower cost, WAAM is applied to fabricate large near-net TC11 components [16].

It is widely accepted that the mechanical properties of alloys are very sensitive to microstructural features [17]. Similar to other additive manufacturing techniques, the repeated melting and solidification processes cause a large thermal gradient in the deposition direction, resulting in epitaxial nucleation and growth of β-grains; consequently, the microstructure of the TC11 alloy prepared by the WAAM technique consists of coarse columnar β-grains [16,18]. Therefore, the static and dynamic mechanical properties of WAAM TC11 titanium alloy may be somewhat different from those of conventionally processed TC11 titanium alloys. Research has been conducted on the basic mechanical properties of TC11 titanium alloys fabricated by the conventional process. Yan et al. [19] studied the effects of different heat treatment methods on the microstructure and quasi-static tensile mechanical properties of TC11 titanium alloy. Four (4) different microstructures and corresponding static mechanical properties were obtained by different heat treatment processes, which proved that the microstructure has significant effects on the mechanical properties of the material. Huang et al. [20] studied the thermal compression behavior of equiaxed TC11 titanium alloy in the temperature range of 900–1060 °C and the strain rate range of 0.001–10 s^−1^. The effect of microstructure on its flow stress softening and adiabatic shear band stability changes were investigated by means of optical microscopy (OM), scanning electron microscopy (SEM), and X-ray diffraction (XRD). Chen et al. [21] determined the parameters in the J–C constitutive model for TC11 titanium alloy by quasi-static tensile tests and high strain rate dynamic compression tests at different temperatures. Zhang et al. [22] studied the tensile mechanical properties of TC11 titanium alloy in the strain rate range of 1 × 10^−3^ s^−1^~1 × 10^3^ s^−1^ and observed the fracture morphology of the specimen. The results showed that with an increase in strain rate of TC11 titanium alloy, the strain rate sensitivity changed from insensitive to sensitive, the strain hardening rate of the material decreased gradually, and TC11 titanium alloy exhibited ductile fracture mechanism during tensile fracture. However, there are relatively few studies on the dynamic mechanical properties and failure mechanisms of TC11 titanium alloy materials prepared by additive manufacturing technology. Zhou et al. [23] studied the microstructure characteristics and mechanical properties of TC11 titanium alloy deposited by laser melting. They found that the microstructure of the deposited sample was composed of coarse columnar grains and equiaxed grains. The mechanical properties also had significant anisotropy characteristics at room temperature. After annealing at 950 °C for 1 h and 550 °C for 2 h, the continuous α phase at grain boundaries was almost completely broken and the distribution of α + β basket-weaved microstructure was more uniform. At room temperature, the anisotropy of mechanical properties was completely eliminated, and the plasticity was enhanced. Zhu et al. [24] prepared a TC11 titanium alloy plate by laser melting deposition process. The morphology, microstructure, and formation mechanism of its grains were systematically studied. Through tensile testing, the LMDed material was found to have high strength and low ductility when compared with traditional forging material. Zong et al. [25] studied laser additive manufacturing TC11 titanium alloy and found that the titanium alloy prepared by laser melting was composed of coarse columnar crystals and fine equiaxed crystals. After annealing at 1263 K for 1 h and 803 K for 6 h, the microstructure showed a basket-weave microstructure. TC11 before and after heat treatment was dynamically compressed at 2800 s^−1^ and 2900 s^−1^ strain rates. TC11 without heat treatment had higher yield strength and poor plasticity. After heat treatment, the yield strength of TC11 titanium alloy decreased, but the plasticity improved. Although numerous scholars have conducted extensive research on the static and dynamic mechanical properties and constitutive relationships of TC11 titanium alloys manufactured by conventional preparation processes and achieved certain results, there are few studies on the static and dynamic mechanical properties and failure mechanisms of TC11 titanium alloy manufactured using WAAM technology, and the plastic constitutive model of WAAM TC11 is rarely studied. Therefore, it is necessary to study the mechanical properties and failure modes of WAAM TC11 titanium alloy.

In this work, the quasi-static and dynamic mechanical properties of TC11 titanium alloy fabricated by WAAM technology under two loading directions (deposition direction and scanning direction) were investigated systematically. Combined with OM and SEM observation of the microstructure and fracture morphology of materials, the failure mechanism of materials was studied. The failure mechanism of the material was studied by observing the microstructure and fracture morphology by OM and SEM. A modified Johnson–Cook (JC) constitutive model considering strain rate sensitivity and the effect of strain rate on strain hardening was established, and the parameters were determined using a Multiple Population Genetic Algorithm (MPGA). This model provides a reference for the numerical simulation of TC11 titanium alloy fabricated by WAAM technology and provides a theoretical foundation for the application of arc additive manufacturing technology in the manufacturing of large complex titanium alloy structural parts.

## 2. Materials and Methods

### 2.1. Materials

A thick plate (180 mm × 30 mm × 200 mm) of TC11 titanium alloy was prepared using WAAM technology, and the material is hereinafter called WAAM TC11 titanium alloy. The instrument used for the preparation of the material is independently developed by the School of Mechanical and Vehicle Engineering of Beijing Institute of Technology. The processing schematic and experimental equipment are shown in Figure 1. TC11 titanium alloy wire with a diameter of 1.6 mm was used as the additive material, and its specific chemical composition is given in Table 1. According to the literature, the microstructure of the as-deposited titanium alloy prepared by WAAM technology features coarse columnar β grain throughout multiple layers, residual stress, and a rough surface [15,18]. Therefore, in order to eliminate the residual strain and obtain the uniform microstructure of TC11 titanium alloy, the material was heat-treated by annealing at 1263 K for 1 h, followed by annealing at 803 K for 6 h [25].

The mechanical properties of WAAM TC11 titanium alloy were investigated via uniaxial quasi-static and dynamic compression experiments. According to GB/T 7314-2017 quasi-static room temperature compression method for metallic materials, a cylindrical compression specimen with a diameter of 4 mm and a length of 4 mm was designed. Given the relationship between the stress balance and uniformity assumptions along the length of the specimen and the size effect in the SHPB experiment, the dynamic compression specimens share the dimensions with the quasi-static compression specimens. Higher loading strain rate can be obtained by using this sample on existing SHPB experimental equipment. The specimen geometry is presented in Figure 2. Considering that the mechanical properties of the heat-treated WAAM TC11 titanium alloy may still differ in the scanning and deposition directions, two sets of specimens with the same geometry and different force directions were prepared. The schematic diagram of the sampling form for specimens is shown in Figure 3. The specimens normal to the circular cross-section along the Z-axis (deposition direction) are the Z-set and the specimens normal to the circular cross-section along the X-axis (scanning direction) are the X-set.

### 2.2. Methods

Quasi-static uniaxial compression experiments on WAAM TC11 titanium alloy were carried out using the Instron 8801 universal materials testing machine. The mechanical properties and failure modes of the material were obtained at different strain rates of 0.001 s^−1^, 0.01 s^−1^, and 0.1 s^−1^ by adjusting the loading speed in the experiments. For quasi-static uniaxial compression experiments, the strain rate, engineering stress, and engineering strain can be obtained from Equation (1). Here,$\;\dot{\varepsilon }\left(t\right)$ is the strain rate, *V*(*t*) is the loading velocity, *l*_0_ is the length of the specimen, *A*_0_ is the initial cross-sectional area of the specimen, and *F*(*t*) and *U*(*t*) are the force on the specimen and the deformation of the specimen at different moments obtained by the universal test machine, respectively.
(1)$$\{\begin{array}{c} \dot{\varepsilon }\left(t\right)=\frac{V\left(t\right)}{l_{0}}\\ \sigma \left(t\right)=\frac{F\left(t\right)}{A_{0}}\\ \varepsilon \left(t\right)=\frac{U\left(t\right)}{l_{0}} \end{array}$$

Under dynamic loading, the tests were carried out using the SHPB system at different strain rates ranging from 700 s^−1^–4000 s^−1^. In this work, the SHPB experimental setup consists of a striker bar, an incident bar, and a transmitter bar made of 19 mm diameter 18 Ni steel, with an impact bar length of 220 mm and an incidence and transmission bar length of 1200 mm. Further, a stress reversal technique [27,28,29,30] is adopted here to avoid reloading of the specimen by reflected stress waves, as shown in Figure 4. 

The SHPB experimental technique is based on two basic assumptions: the one-dimensional stress assumption and the homogeneity assumption. Applying the one-dimensional stress wave theory, the force–displacement curve at the interface between the compression bar and the specimen can be obtained from the strain signal. Based on the homogeneity assumption, the loading strain rate, strain, and stress of the material can be determined, which can be given by Equation (2). The specific theoretical calculation procedure for the SHPB can be found in Ref. [31].
(2)$$\{\begin{array}{c} \dot{\varepsilon }\left(t\right)=-2\frac{C}{l_{s}}\varepsilon_{r}\left(t\right)\\ \varepsilon \left(t\right)=-2\frac{C}{l_{s}}\int_{0}^{t}\varepsilon_{r}\left(t\right)dt\\ \sigma \left(t\right)=\frac{A_{bar}}{A_{s}}E\varepsilon_{t}\left(t\right) \end{array}$$
where $\varepsilon_{r}\left(t\right)$ and $\varepsilon_{t}\left(t\right)$ are the reflected and transmitted strains measured by strain gauges on the compression bars, respectively. *E*, *C*, and *A_bar_* are the modulus of elasticity, elastic wave velocity, and cross-sectional area of the incident and transmitted bars, respectively. *A_s_* and *l_s_* are the cross-sectional area and length of the specimen, respectively. In addition, in uniaxial compression experiments, engineering strains and stresses can be converted to real strains and stresses using Equation (3).
(3)$$\{\begin{array}{c} \varepsilon_{T}=-ln(1-\varepsilon )\\ \sigma_{T}=\sigma \left(1-\varepsilon \right) \end{array}$$

The original specimens and deformed specimens were axially sectioned parallel to the compression axis and the cut surface was prepared for optical microstructure examination using standard polishing and etching techniques. OM observations were carried out on an OLYMPUS BX51M optical microscope made in China. In addition, the fracture morphologies of the specimens were observed by Quanta 250FEG (FEI, Hillsboro, OR, USA) field emission environmental scanning electron microscope. 

The experimental test workflow is given in Figure 5. Firstly, the TC11 titanium alloy fabricated by wire arc additive manufacturing was wire cut to obtain the experimental specimens, then the Instron 8801 universal materials testing machine and SHPB experimental device were used to load the experimental specimens quasi-statically and dynamically, respectively, and finally the microstructure and fracture morphology of the specimens were observed by OM and SEM.

## 3. Results and Discussion

### 3.1. Deformation of Specimens

Macroscopic deformations of WAAM TC11 titanium alloy specimens after static and dynamic compression are shown in Figure 6. When the direction deposition and the direction scanning sample are damaged, the angle between the crack or section and the sample axis (loading direction) is approximately 45° under quasi-static loading conditions. Under dynamic loading conditions, both deposited and scanned specimens were deformed to varying degrees before damage. When loaded in the deposited direction, damage occurs when the loading strain rate is higher than 3000 s^−1^. However, when loaded in the scanned direction, damage occurred when the loading strain rate was above 4000 s^−1^ and the angle between the fracture and the specimen axis (loading direction) was approximately 45°.

### 3.2. Analysis of Mechanical Properties in the Deposition and Scanning Directions

Numerous research results show that the microstructure will directly affect the mechanical properties of titanium alloys [6]. The microscopic morphology of the longitudinal sections of the X and Z cylindrical compression specimens were observed by OM. The microstructure morphology of the corresponding specimens was obtained as shown in Figure 7. From Figure 7, the original β grain is broken to varying degrees and the lamellar α phase is staggered inside the β phase grains and woven into a net basket, which is a typical basket-weave microstructure in both scanning direction and deposition direction. However, as you can see in Figure 7a, there are several α phases that are more than 100 µm long, while in Figure 7b such α phases are not visible. These differences may result in inconsistent mechanical properties of materials in both directions.

The engineering stress-strain curves of WAAM TC11 titanium alloy under different loading directions were obtained by quasi-static and dynamic compression experiments. The real stress-strain curve of the material was obtained by using Equation (3). The true stress-true strain curves for the same strain rate of two sets of specimens under quasi-static loading conditions are given separately in Figure 8. W-Z and W-X in the legend denote WAAM-TC11 specimens subjected to compression in the direction parallel to the deposition direction and the scanning direction, respectively, with the numbers representing different specimens. It can be seen from Figure 8 that within the strain rate range of 0.001 s^−1^–0.1 s^−1^, both groups of WAAM TC11 titanium alloy samples underwent elastic stage, yield stage, strengthening stage, and failure stage. The elastic stage, yield stage, and strengthening stage of the two groups of samples coincided well, but there was no obvious yield platform in the yield stage. Therefore, the yield strength of materials under quasi-static condition is defined as the stress value corresponding to 0.2% plastic strain. Under the same strain rate, the compressive strength of the scanning direction was slightly larger than that of the deposition direction, but the difference was less than 4%. The failure strain in the scanning direction was greater than that in the deposition direction.

Figure 9 shows the real stress-strain curves of the two groups of samples at the same strain rate under dynamic loading conditions. It can be seen from the Figure 9 that the yield strength and flow stress of the two groups of samples at the same strain rate have no obvious difference and the mechanical properties do not show anisotropy. With the increase in strain rate, the yield strength and flow stress of the two groups of samples show a positive correlation of strain rate. A comparison of the stress-strain curves of LAMed TC11 titanium alloy at 2900 s^−1^ strain rate with WAAM TC11 titanium alloy at 3000 s^−1^ strain rate is given in Figure 9e. From the figure, it can be concluded that although the yield strengths of TC11 titanium alloys prepared by different processes are close at similar strain rates, the strain hardening trends are different, further illustrating the need to study the mechanical properties of WAAM TC11 titanium alloy. It can be observed in Figure 9e,f that the specimen in the deposition direction has fractured at strain rates of 3000 s^−1^ and 3500 s^−1^, while the specimen in the scanning direction only has plastic deformation without fracture, which proves that the plasticity of the specimen in the scanning direction is superior to that in the deposition direction from the side.

The study of the deformation and damage of the specimens after the test and the obtained stress–strain curves showed that there was no significant directional difference in the mechanical properties of the elastic and plastic segments of the material in the deposition and scanning directions. However, the plasticity in the scanning direction was better than that in the deposition direction.

### 3.3. Effect of Strain Rate

The true stress-strain curves for the two sets of specimens at different strain rates under quasi-static and dynamic conditions are given in Figure 10 and Figure 11, respectively. The yield strength of the material under quasi-static conditions increases slightly with increasing strain rate and the flow stress changes less as shown in Figure 10. Combining the stress-strain curves under quasi-static conditions in Figure 10, the yield and flow stresses of the material are significantly elevated under dynamic loading conditions as shown in Figure 11.

To study the strain rate effect of the material quantitatively, the yield strength and flow stress at the given plastic strain 0.05 of WAAM TC11 titanium alloy under different strain rates (0.001 s^−1^–3500 s^−1^) were given in Table 2. The results show that the yield strength and the flow stresses increase with the strain rates. Under dynamic conditions, the yield strength and the flow stress rise more quickly. The yield strength and the flow stress do not increase linearly over the range of strain rates studied, showing an inflection point in the tendencies of the flow stress with strain rates. To study the strain rate sensitivity, the strain rate sensitivity coefficient is defined by parameter β [32,33].
(4)$$\beta =\frac{\partial \sigma }{\partial \dot{ln\varepsilon }}=\frac{\sigma_{2}-\sigma_{1}}{ln({\dot{\varepsilon }}_{2}/{\dot{\varepsilon }}_{1})}\beta =\frac{\partial \sigma }{\partial \dot{ln\varepsilon }}=\frac{\sigma_{2}-\sigma_{1}}{ln({\dot{\varepsilon }}_{2}/{\dot{\varepsilon }}_{1})}$$

Here, $\sigma_{1}$ and $\sigma_{2}$ are equivalent stresses at strain rates ${\dot{\varepsilon }}_{1}$ and ${\dot{\varepsilon }}_{2}$, respectively.

The effect of strain rate on the mechanical properties of the material is expressed using the yield strength and flow stress at the given plastic strain of 0.05 versus the strain rate in logarithmic form and the strain rate sensitivities parameters under quasi-static and dynamic loading conditions of the two sets of specimens are labeled in the figure, as shown in Figure 12. The results show that the strain rate sensitivities to yield and flow stresses in the deposited and scanned directions of WAAM TC11 titanium are consistent with the equivalent strain rate. However, the relationship between the yield and flow stresses and the logarithm of the strain rate is not simply linear. With the strain rate sensitivity coefficient at a high strain rate being approximately ten times higher than the strain rate sensitivity coefficient at quasi-static, there is a clear transition from low to high strain rate. This is mainly due to the difference in the thermal activation mechanism under dynamic and static loading, which refers to the energy fluctuation across the energy barrier generated by the thermal movement of atoms under external stress [34]. The relationship between activation energy (ΔG) and strain rate is expressed by [35,36]:(5)$${\dot{\varepsilon }}_{0}=b\rho_{m}d\gamma ,\Delta G=kTln\frac{{\dot{\varepsilon }}_{0}}{\dot{\varepsilon }}$$

Here, ${\dot{\varepsilon }}_{0}$ is the reference strain rate, b is the Burgers vector, $\rho_{m}$ is the mobile dislocation density, d is the moving distance of dislocation to overcome obstacles, γ is the vibrational frequency of dislocation lines, k is the Boltzmann constant, and T is temperature. Accordingly, a higher $\dot{\varepsilon }$ often induces a lower ΔG as there is less time available to overcome the barriers to dislocation movement. Therefore, at lower strain rates, the activation energy is relatively larger, which effectively helps dislocation to overcome obstacles and thus the thermal activation effect is fully utilized, leading to smaller external stress and lower strain rate sensitivity. On the contrary, at a higher strain rate, the activation energy is insufficient to overcome the barrier. The thermal activation effect gradually weakens, leading to the larger external stress and higher strain rate sensitivity. For comparison, the strain rate sensitivity tendency for WAAM TC11 is very similar to the reported TC4 titanium alloy [37].

The difference between the yield strength and the flow stress at an equivalent plastic strain of 0.05 for the same strain rate is given in Table 3 for both groups of specimens. The results show that the absolute error between the yield strength of the two groups of specimens is a minimum of −14.25 MPa and a maximum of 6.67 MPa. The absolute error of the flow stress at an equivalent plastic strain of 0.05 is a minimum of −14.25 MPa and a maximum of 12.28 MPa. These results further prove that there is no significant difference between the mechanical properties of the material in the scanning direction and the deposition direction.

### 3.4. Failure and Microstructural Evolution Analysis

#### 3.4.1. Macro-Failure Analysis

Based on the analysis of the deformation of the specimens after loading in Section 3.1, it can be seen that in the uniaxial compression process, the damage of both sets of specimens of WAAM TC11 titanium alloy occurs in the direction of the maximum shear stress in the plane (the angle with the compression axis is 45°). When plastic deformation of the specimens occurs during loading, circumferential stresses appear on the specimen surface, which induces the specimens to be loaded in tension. Therefore, the compression-shear region and the tension-compression-shear region coexist in the shear damage plane during the whole loading process of the specimen, as shown in Figure 13. Figure 14 gives the damage shape of the cylindrical specimen after uniaxial compression, which is in good agreement with Figure 13.

#### 3.4.2. Micro-Failure Analysis

The specimens were cut along the longitudinal section after static and dynamic loading conditions and the microstructure of the section was observed by OM. The microstructures of typical specimens in the longitudinal sections under quasi-static and dynamic loading conditions are shown in Figure 15 and Figure 16, respectively. In Figure 15, area A shows the specimen section after deformation, and areas B and C show the local magnification at different magnifications, respectively. It can be observed that both α and β phases of the material show different degrees of compression deformation along the loading direction under quasi-static loading conditions. The specimens in the deposition direction did not stop loading in time after the damage of the specimens, resulting in excessive compression of the specimens. The deformation of the α-phase and β-phase was more intense, which led to a more difficult observation of the shear deformation zone. In contrast, cracks, microcracks, and shear deformation bands between two cracks can be observed in the specimen in the scanning direction and the shear deformation bands are parallel to the crack direction and at an angle of approximately 45° to the loading direction. The microstructure under dynamic loading conditions is shown in Figure 16, where it can be observed that the α-phase and β-phase deformation areas are small and concentrated. Although no complete adiabatic shear band is observed due to the complete fracture of the specimens, the crack extension along the adiabatic shear band can be observed in Figure 16a,b, proving that the fracture failure mechanism of WAAM TC11 titanium alloy is an adiabatic shear deformation fracture.

The fractures of the fractured specimens were observed using SEM, and the typical fracture morphologies of the specimens in the deposition and scanning directions were obtained, as shown in Figure 17 and Figure 18, respectively. The loading strain rates were 3500 s^−1^ and 4000 s^−1^ for the two specimens, and the overall fracture morphology and the local fracture morphology at different magnifications are presented in the two sets of figures, respectively. It can be seen from (a) of the two groups of figures that the microscopic fracture morphology of WAAM TC11 is mainly composed of dimple zone and smooth zone. Among these, the dimples are mainly distributed on the left and right sides and the lower part of the fracture, and the smooth area is mainly distributed on the middle and upper part of the fracture, and the dimples are relatively large in the scanning direction samples. In the two figures, (b) and (c) present the microscopic morphologies of dimples with different magnification ratios. Under high strain rate loading, the samples in both groups form typical parabolic dimples, which belong to shear fracture mechanism. By comparing the samples in two groups, it can be determined that the dimples formed by the samples in the scanning direction after fracture have larger overall deformation and more obvious elongation. Combined with the ratio of dimple area and dimple morphology, the plasticity of scanning direction is better than that of deposition direction, which is consistent with the results of sample deformation in Section 3.1 and stress-strain curve in Section 3.2.

## 4. The Constitutive Analysis

### 4.1. Parameter Fitting and Applicability Analysis

The Johnson–Cook (J–C) model is an empirical viscoplastic constitutive model that can better describe the strain rate effect, work-hardening effect, and temperature softening effect of metal materials. Based on the previous analysis, there is no significant difference in the mechanical properties of the material in the plastic section. Without considering the influence of the material processing direction, the J–C model can be used to describe the dynamic plastic mechanics constitutive relationship of WAAM TC11 titanium alloy. In the J–C model, the flow stress σ can be expressed as:(6)$$\sigma =\left(A+B\varepsilon^{n}\right)\left(1+Cln{\dot{\varepsilon }}^{*}\right)\left(1-T^{*m}\right)$$
where *A*, *B*, *n*, *C*, and *m* are coefficients to be determined based on experimental data. *A* is the yield strength at reference temperature and reference strain rate; *B* is the strain hardening coefficient; *n* is the strain hardening index; *C* is the strain rate hardening coefficient; *m* is the thermal softening index; *ε* is the equivalent plastic strain ${\overset{•}{\varepsilon }}^{*}$ is the strain rate. $T^{*}$ is the relative temperature. The experiments at room temperature do not consider the effect of the temperature term and take 0.001 s^−1^ at room temperature as the reference strain rate. Combining the experimental data at different strain rates, the final values of *A*, *B*, *n*, and *C* are 1046.62 MPa, 979.41 MPa, 0.7070, and 0.0146, respectively. Following that, the J–C model takes the following form:(7)$$\sigma =\left(1046.62+979.41\varepsilon^{0.707}\right)\left(1+0.0146{\dot{ln\varepsilon }}^{*}\right)$$

The comparison between flow stress in the direction of material deposition and J-C constitutive fitting results in the experiment is shown in Figure 19. It can be seen from the figure that the fitting results are in good agreement with the experimental results under quasi-static conditions, but at high strain rates, the yield strength of the material changes little with the increase in strain rate and the error with experimental results is large. Moreover, with the increase in strain, the flow stress obtained by J-C fitting is higher than the experimental value. The study of the characteristics of the J-C model by LIANG [38] and XU [39] shows that the conventional J-C model cannot characterize the mechanical behavior of the materials with large deviations in strain rate sensitivity coefficients under static and dynamic loading conditions. The materials either decrease strain hardening rate with increasing loading strain rate or remains constant. Therefore, the conventional J-C model cannot accurately characterize the plastic flow behavior of WAAM TC11 titanium alloy and a modification of the constitutive model is required.

### 4.2. Model Modification and Parameter Fitting

As mentioned above, the compressive mechanical behavior of WAAM TC11 shows an obvious strain rate correlation. However, the sensitivity of yield strength and strain hardening behavior to strain rate is different. With an increase in strain rate, the yield strength of materials changes from insensitive to sensitive and the strain hardening effect decreases. In view of the above analysis, the J–C constitutive model is modified. Two parameters, *C*_1_ and *C*_2,_ were used to characterize the influence of strain rate on yield behavior and strain hardening behavior, respectively. *C*_1_ is a function related to strain rate, which can be characterized as $a\dot{\varepsilon }+b$. The transition strain rate ${\dot{\varepsilon }}_{T}$ was introduced to characterize the transition phenomenon of strain rate sensitivity of flow stress from low strain rate region to high strain rate region. The modified J–C constitutive model (MJ–C constitutive model) is shown in Equation (8):(8)$$\sigma \;=A\left(1+C_{1}{\dot{ln\varepsilon }}^{*}\right)+B\varepsilon^{n}\left(1+C_{2}{\dot{ln\varepsilon }}^{*}\right),\;{\dot{\varepsilon }}^{*}=\frac{\dot{\varepsilon }+{\dot{\varepsilon }}_{T}}{{\dot{\varepsilon }}_{0}}$$

Genetic algorithms can be used to calibrate the parameters of the complex forms of material constitutive and equation of state. At present, this method has been applied to calibrate the parameters of the explosive state equation [40,41] and the material constitutive model [42,43,44]. The Multiple Population Genetic Algorithm (MPGA) used in this work not only inherits many of the advantages from the standard genetic algorithm (SGA), such as no need for gradient information, strong robustness, good ability to capture global optima, etc. [45]. Furthermore, the strategy of multi-population co-evolution can obtain higher accuracy, faster convergence rate, and effectively overcome the problem of premature convergence [46]. In this work, the MPGA is briefly described as follows: a number of initial subpopulations are randomly generated within the solution range. The reciprocal of the absolute value of the error in the flow stress obtained by Equation (8) and the experiment is used as the fitness function. Selection, crossover, mutation migration etc. are performed within each generation. The calculation is cyclic until the maximum number of generations of population evolution is reached or the optimal individuals remain constant for 50 consecutive generations. The optimal individuals are output at the end of the calculation, which is the value of the parameter obtained. The specific calculation process is shown in Figure 20, where *j* is the number of generations; *J* is the maximum number of generations, 300; *N* is the number of subpopulations, ten (10). *M* is the number of individuals in the subpopulation, 100; *P*(*j*,*n*) is the nth subpopulation in the jth generation. *F* is the fitness of the current optimal individual. *i* is the number of generations for which the current optimal individual is maintained. *E*_max_ is the maximum number of generations for which the optimal individual is maintained, 50. The range of values of the parameters is given in Table 4. The calculated final parameters are shown in Table 5. 

The comparison between flow stress in the direction of material deposition and MJ–C constitutive fitting results is shown in Figure 21. The modified constitutive model is in good agreement with the experimental stress-strain curves at different strain rates. Figure 22 compares the flow stress in the scanning direction of the material in the experiment with the fitting results of MJ–C constitutive model, which also shows good fitting results. Therefore, considering the strain rate sensitivity and the effect of strain rate on strain hardening effect, MJ–C constitutive model can well describe the static and dynamic mechanical properties of WAAM TC11 titanium alloy.

There are several focuses of attention to be conveyed regarding the limitations in our study. First, due to the limitation of the experimental equipment, experimental data in the range of medium strain rate (1 s^−1^–10^2^ s^−1^) are missing. Second, the experiments were carried out at room temperature and not at different temperatures, making MJ–C constitutive mode only applicable to room temperature conditions. Finally, the obtained MJ–C constitutive model needs to be redeveloped before it can be used in numerical simulation.

## 5. Conclusions

To investigate the mechanical properties and failure modes in the deposition and scanning directions, quasi-static and dynamic compression experiments were carried out on heat-treated WAAM TC11 titanium alloy. The material and evolution of the microstructure and the failure mode of the specimens after fracture were investigated through metallographic observations and fracture morphology analysis. A modified MJ–C intrinsic structure model for WAAM TC11 was developed by collating and analyzing the experimental data. The main conclusions obtained are as follows.

In the strain rate range of 0.001 s^−1^–4000 s^−1^, the yield strength and flow stress of the material in the scanning and deposition directions do not differ significantly and exhibit strain rate strengthening effects, but the strain rate sensitivity of the material under quasi-static loading conditions is much less than under dynamic loading conditions. The specimens which were fractured all broke along the maximum shear plane, with the plane of fracture at approximately 45° to the loading direction, and the crack extension along the adiabatic shear band could be observed only under dynamic loading. In addition, the fracture of both deposited and scanned specimens under dynamic loading consisted of a dimpled area and a smooth area, and the dimpled areas exhibited parabolic shear dimples. However, in the scanning direction sample, the dimples proportion was larger at the fracture and the dimples were more obviously elongated and deformed, proving that the scanning direction has better plasticity. Based on the experimental results, the MJ–C constitutive model was obtained by considering the strain rate sensitivity and the effect of strain rate on strain hardening. The accuracy and reasonableness of the MJ–C constitutive model was proven by comparison with the experimental results. Based on the research results of this paper, further research will be carried out on the mechanical properties of WAAM TC11 titanium alloy considering the effect of temperature and stress state.

## Figures and Tables

**Figure 1 materials-15-03917-f001:**
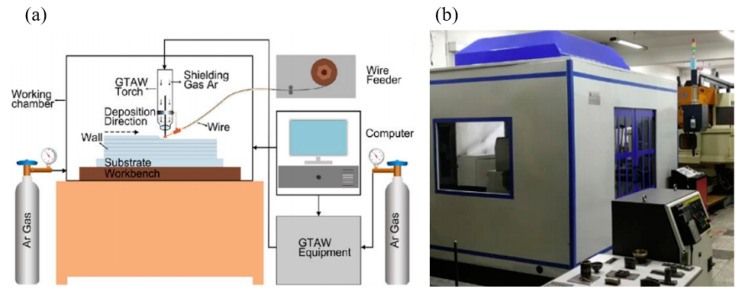
(**a**) Schematic diagram of WAAM technology [26]; (**b**) WAAM equipment.

**Figure 2 materials-15-03917-f002:**
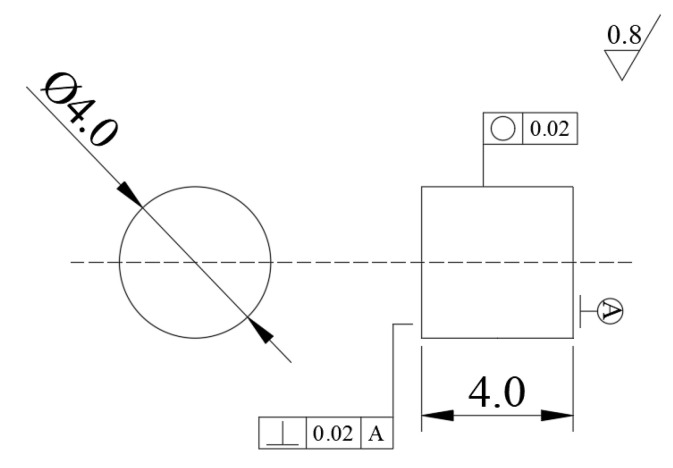
Schematics of specimens.

**Figure 3 materials-15-03917-f003:**
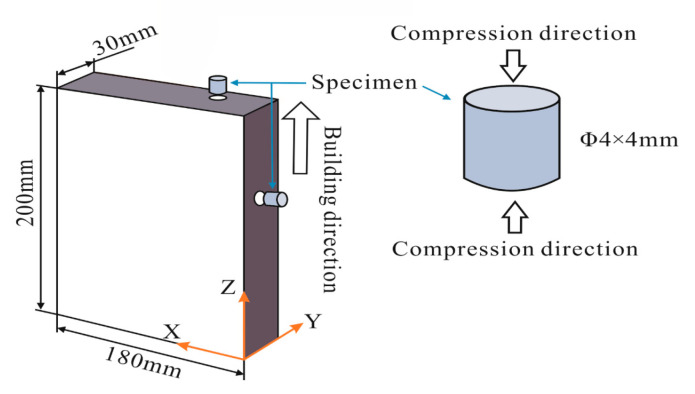
Schematic diagram of the sampling method for specimens.

**Figure 4 materials-15-03917-f004:**
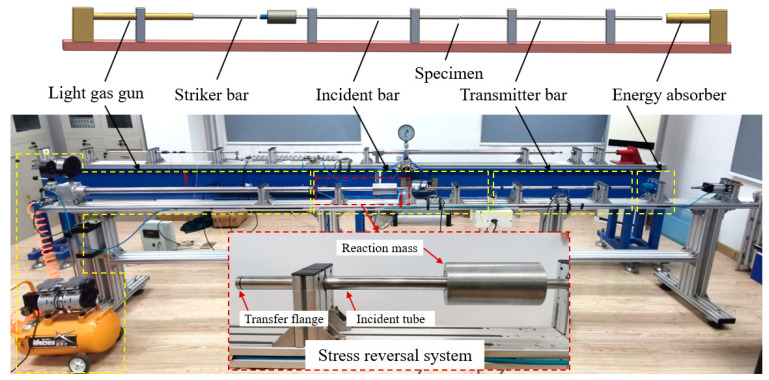
Schematic diagram of the SHPB experimental setup.

**Figure 5 materials-15-03917-f005:**
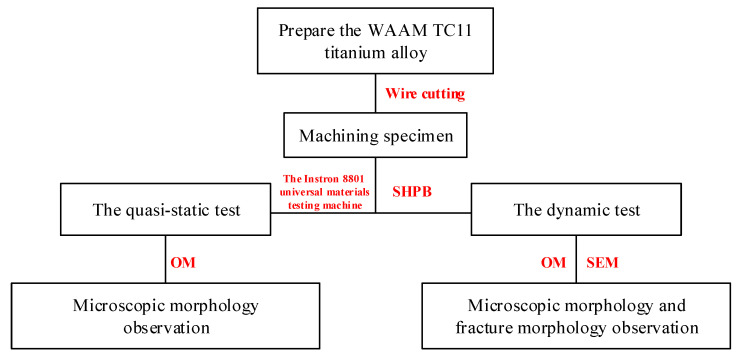
Schematic diagram of the workflow of experimental testing.

**Figure 6 materials-15-03917-f006:**
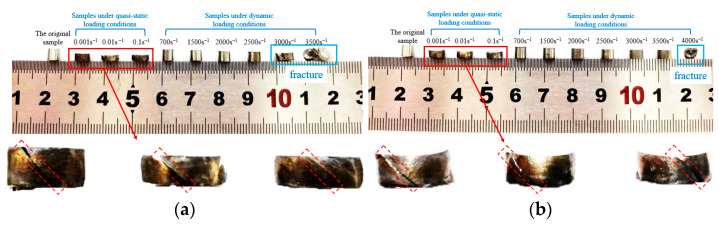
Macro-morphology of uniaxially compressed specimens after compression. (**a**) Group Z specimens (deposition direction). (**b**) Group X specimens (scanning direction).

**Figure 7 materials-15-03917-f007:**
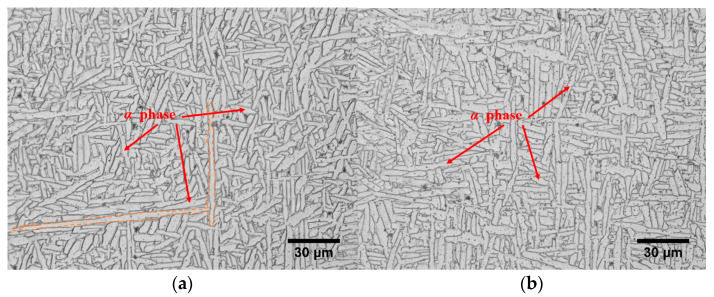
Microstructure of the longitudinal sections of the two groups of samples. (**a**) Group Z specimens (deposition direction). (**b**) Group X specimens (scanning direction).

**Figure 8 materials-15-03917-f008:**
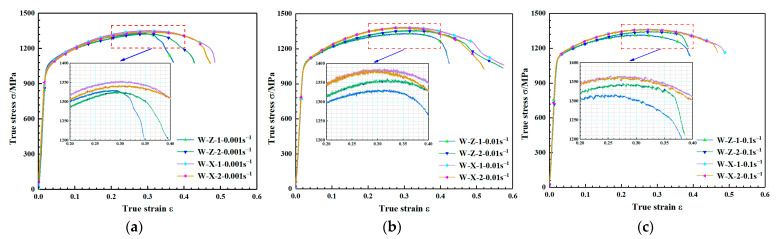
Stress–strain curves for different specimens at the same strain rate under quasi-static loading conditions. (**a**) Strain rate of 0.001 s^−1^. (**b**) Strain rate of 0.01 s^−1^. (**c**) Strain rate of 0.1 s^−1^.

**Figure 9 materials-15-03917-f009:**
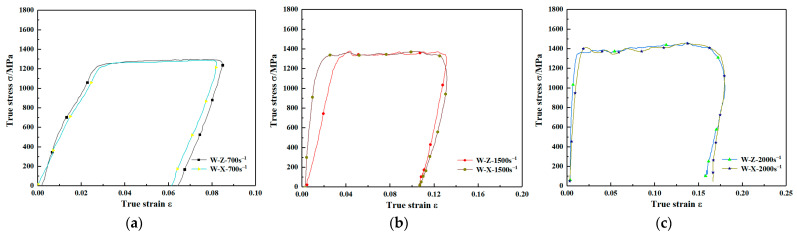
Stress–strain curves for different specimens at the same strain rate under dynamic loading conditions. (**a**) Strain rate of 700 s^−1^. (**b**) Strain rate of 1500 s^−1^. (**c**) Strain rate of 2000 s^−1^. (**d**) Strain rate of 2500 s^−1^. (**e**) Strain rate of 3000 s^−1^. (**f**) Strain rate of 3500 s^−1^.

**Figure 10 materials-15-03917-f010:**
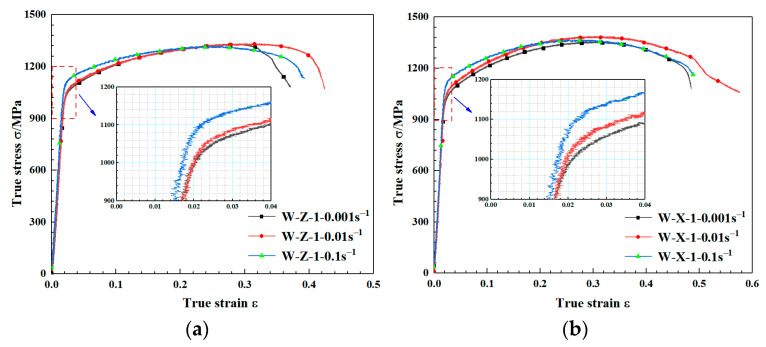
Stress–strain curves for the same specimen at different strain rates. (**a**) Group Z specimens (deposition direction). (**b**) Group X specimens (scanning direction).

**Figure 11 materials-15-03917-f011:**
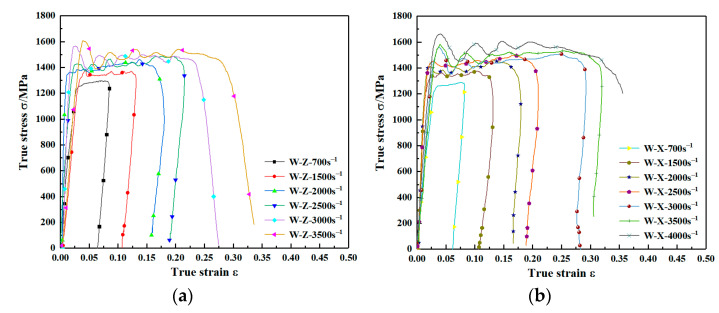
Dynamic stress–strain curve for WAAM TC11 titanium alloy. (**a**) Group Z specimens (deposition direction). (**b**) Group X specimens (scanning direction).

**Figure 12 materials-15-03917-f012:**
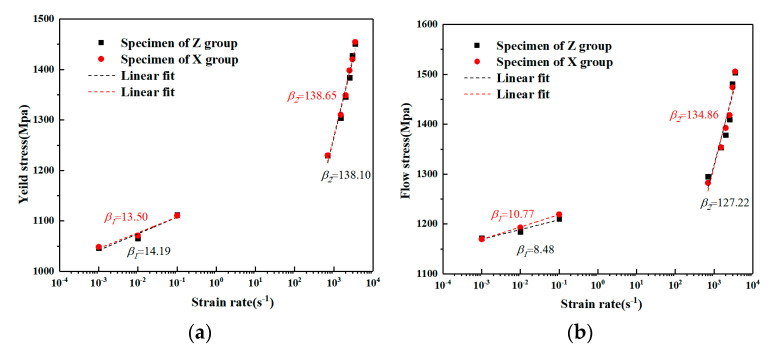
Strain rate sensitivity of yield and flow stresses under different loading conditions. (**a**) Yield strength. (**b**) Flow stress at an equivalent plastic strain of 0.05.

**Figure 13 materials-15-03917-f013:**
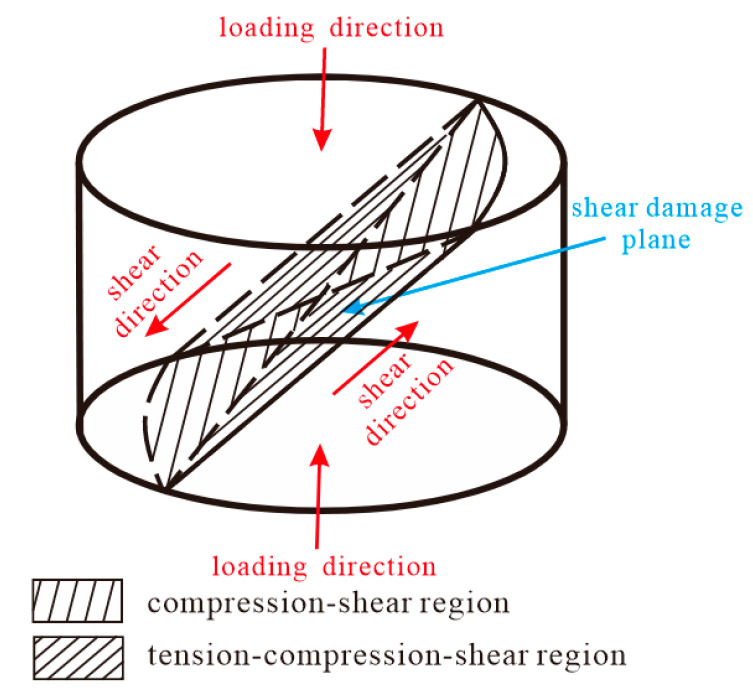
Schematic diagram of the compression instability of the sample.

**Figure 14 materials-15-03917-f014:**
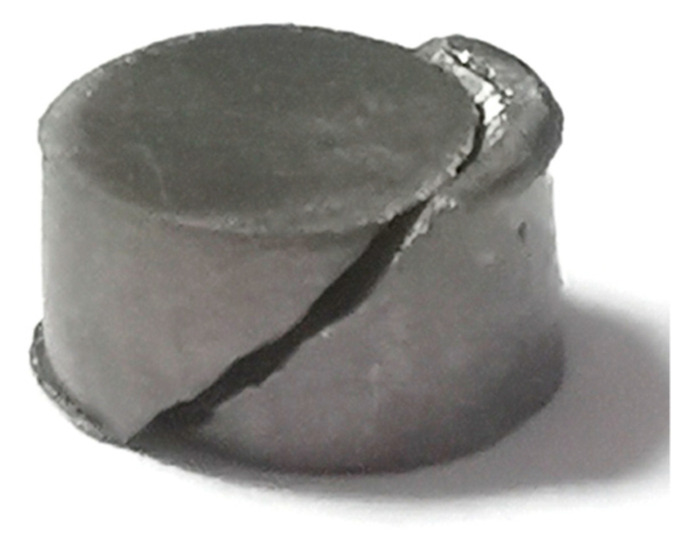
Macroscopic damage profile of compressed specimens.

**Figure 15 materials-15-03917-f015:**
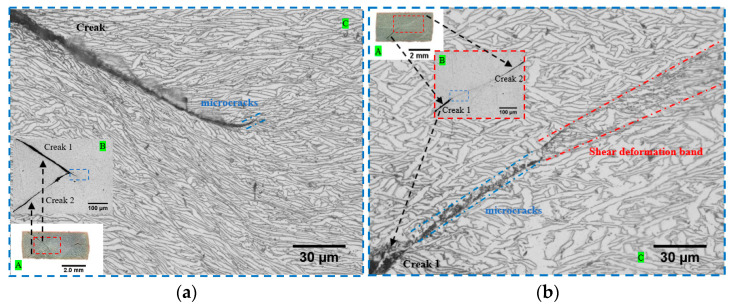
Microstructure of the specimen in longitudinal section under quasi-static conditions. (**a**) W-Z-0.001 s^−1^. (**b**) W-X-0.001 s^−1^. Region A is the cross-sectional photograph of the specimen after deformation, region B is the local magnified microstructure within the red-framed area in region A, and region C is the local magnified microstructure within the blue-framed area in region B.

**Figure 16 materials-15-03917-f016:**
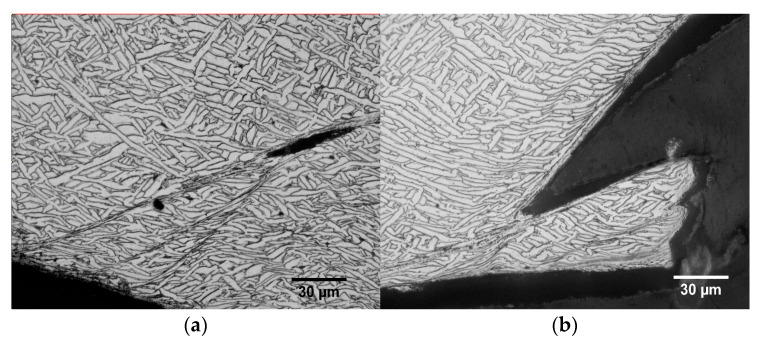
Microstructure of the specimen in longitudinal section under dynamic. (**a**) W-Z-3500 s^−1^. (**b**) W-X-4000 s^−1^.

**Figure 17 materials-15-03917-f017:**
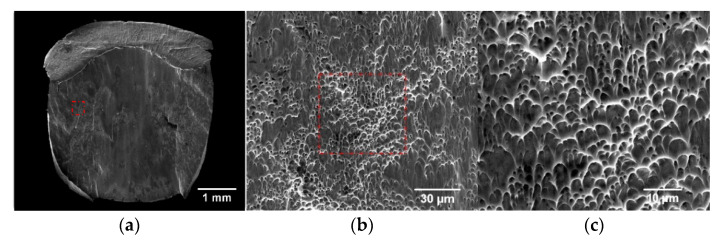
Fracture morphologies of specimen in the deposition direction. (**a**) Full view of the fracture. (**b**) 2000× magnification. (**c**) 5000× magnification.

**Figure 18 materials-15-03917-f018:**
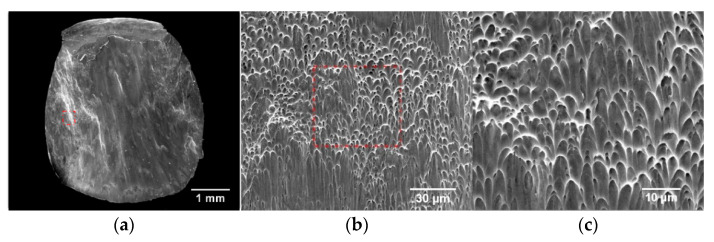
Fracture morphologies of specimen in scanning direction. (**a**) Full view of the fracture. (**b**) 2000× magnification. (**c**) 5000× magnification.

**Figure 19 materials-15-03917-f019:**
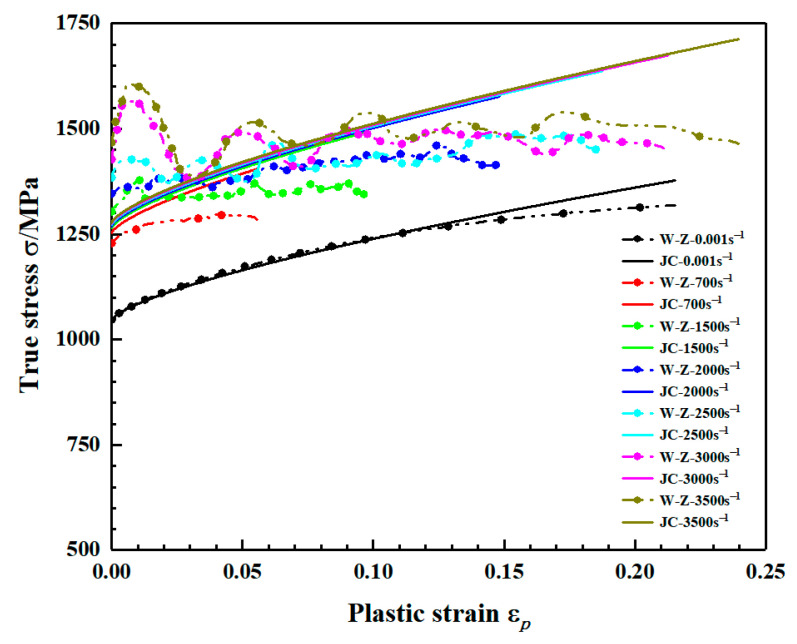
Comparison of flow stresses in the J–C model and experiments (deposition direction).

**Figure 20 materials-15-03917-f020:**
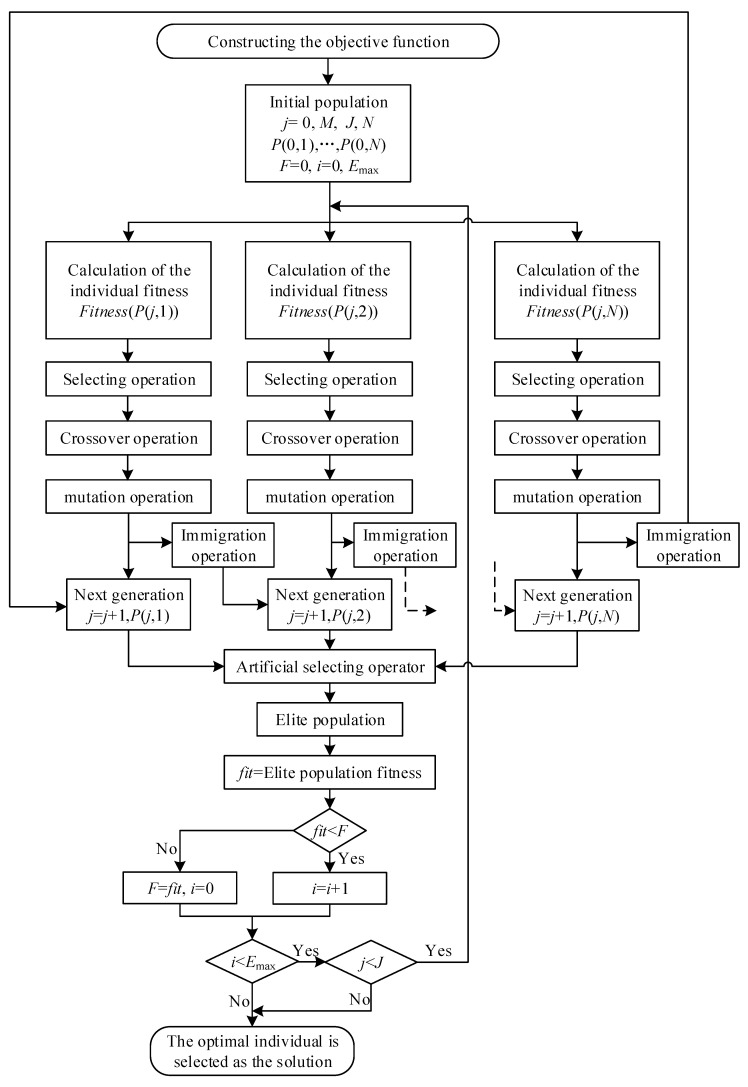
Schematic diagram of MPGA.

**Figure 21 materials-15-03917-f021:**
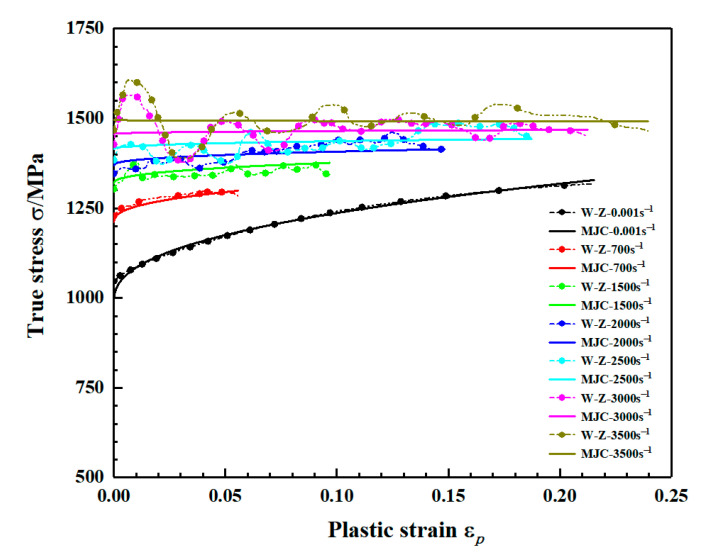
Comparison of flow stresses in the MJ–C model and experiments (deposition direction).

**Figure 22 materials-15-03917-f022:**
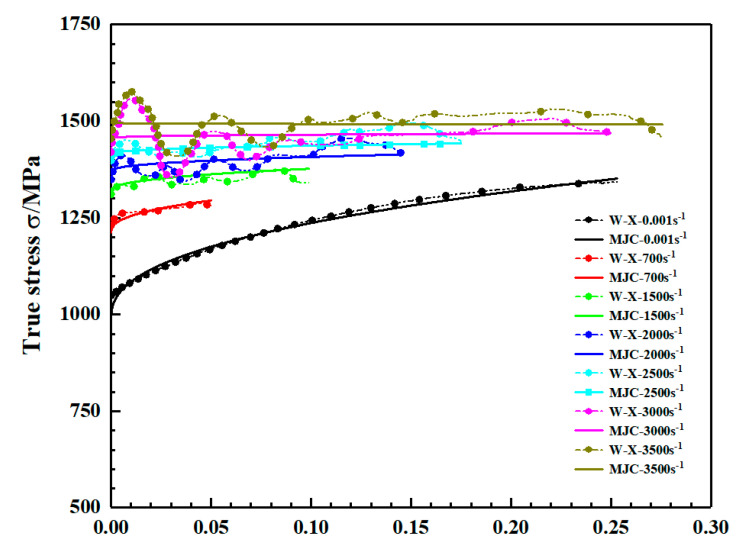
Comparison of flow stresses in the MJ–C model and experiments (scanning direction).

**Table 1 materials-15-03917-t001:** Specific chemical composition of TC11 titanium alloy wire.

Al	Mo	Zr	Si	Ti	Impurities
5.8~7.0	2.8~3.8	0.8~2.0	0.2~0.35	Residuals	≤1.042

**Table 2 materials-15-03917-t002:** Yield strength and flow stress of materials at different strain rates.

Group	Strain Rate/s^−1^	0.001	0.01	0.1	700	1500	2000	2500	3000	3500
Z	*σ_s_*/MPa	1046.76	1065.15	1112.13	1229.61	1303.74	1346.22	1384.05	1427.05	1450.76
	*σ*_0.05_/MPa	1171.78	1184.38	1210.84	1294.83	1353.12	1378.41	1409.66	1480.81	1503.72
X	*σ*_s_/MPa	1048.48	1071.12	1110.64	1229.83	1310.56	1348.96	1398.30	1420.38	1454.56
	*σ*_0.05_/MPa	1169.79	1193.14	1219.42	1282.55	1354.04	1392.66	1418.45	1474.01	1505.57

**Table 3 materials-15-03917-t003:** Absolute errors of yield strength and flow stress in both loading directions for the same strain rate.

Strain Rate/s^−1^		0.001	0.01	0.1	700	1500	2000	2500	3000	3500
Absolute Error	*σ_s_*	−1.72	−5.97	1.49	−0.22	−6.82	−2.74	−14.25	6.67	−3.80
	*σ* _0.05_	1.99	−8.76	−8.58	12.28	−0.92	−14.25	−8.97	6.80	−1.85

**Table 4 materials-15-03917-t004:** Range of parameter values.

Parameters	*A*/MPa	*a*	*b*	*B*/MPA	*n*	*C* _2_	${\dot{\varepsilon }}_{T}\;$
Range of values	[400,1000]	[0,1]	[0,1]	[500,200]	[0,1]	[−1,1]	[0,1000]

**Table 5 materials-15-03917-t005:** MJ–C fitting parameters.

*A*/MPa	*a*	*b*	*B*/MPA	*n*	*C* _2_	${\dot{\varepsilon }}_{T}\;$
441.63	1.14 × 10^−5^	0.2512	1605.9	0.4083	−0.01225	130

## Data Availability

The data presented in this study are available on request from the corresponding author.

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
