# Peer review of "Dynamic Mechanical Properties of TC11 Titanium Alloys Fabricated by Wire Arc Additive Manufacturing"

_materials, 2022, doi:10.3390/ma15113917_

Round 1
Reviewer 1 Report
The quality of the conducted study is high. The presented results of the performed experiments have scientific and practical meaning. A literary review has been made quite fully. In general, the research topic is relevant.
In order for the article to be better perceived by readers, we advise the authors to refine the article a little.
We suggest you supplement the section “Materials and Methods” and expand it with more details. Describe the equipment used, microscopes, ... Under what conditions and on what equipment the research was carried out. Specify the equipment brand and manufacturer in parentheses.
In Figure 5 there are Chinese hieroglyph, it is better to translate them into English. Also, the drawings of the destroyed samples at the bottom of Fig.5 are quite small and difficult to consider. It would be good to increase them a little.
The authors write: “microstructure morphology of WAAM TC11 titanium alloy 201 in scanning direction and deposition direction are similar” (lines 201-202). But at the same time, in Figure 6a there are several crystals with a length of more than 100 microns, and in Figure 6b such crystals are not visible. This is quite a significant difference in the structure. You should choose clearer photos where there will be no such difference.
On lines 220-221 you write that the deformation was different for different samples. But at the beginning of the article you write that the properties strongly depend on the microstructure and then show that the microstructure of different samples is identical. And the properties are different. Although the photos show differences in the microstructure, as I wrote earlier. I think you should build a unified system. If you write about the identity of the structure, then show the same photos and write that the changes in properties are within the measurement error.
In Figure 12, it would be good to translate the entire text into English.
Also, for the experiment results in Figures 10 and 11, please, show the confidence interval. This would be useful, since the values for the samples obtained do not differ much.
Reviewer 2 Report
It is remarkable to study the dynamic properties of the special alloy fabricated by an advanced method. Some comments are below.
1) At first, I mistook as all specimens were tested in the SHPB experimental setup. Furthermore all tests were carried on until the fracture of the specimens. Please, revise the manuscript to show clearly the kinds of tests and whether the specimens were fractured or not.
2) All specimens were shear-fractured in the quasi-static test. On the other hand, most specimens were not fractured in the SHPB test. In other words, the tests were not carried on until the fracture in the SHPB test. It resulted that the specimens were unloaded during the strain hardening stage. I am afraid that the early unloading may give wrong information on the stress level. Why did these incompleteness happen? Is there no way to solve these problems?
3) The dynamic strain rate ranging from 700 to 3500 s-1 were kept constant during loading? If it were, how do you know it?
4) For σ1 and σ2 in equation (4), did you choose the stress at what true strain level?
Reviewer 3 Report
Dear Tian et al.,
The manuscript “Dynamic Mechanical Properties of TC11 Titanium Alloys Fabricated by Wire Arc Additive Manufacturing” (materials-1733265) by Tian et al. provide a theoretical foundation for the application of arc additive manufacturing technology in the manufacturing of large complex titanium alloy structural parts. The topic is interesting, but I think this article should reconsider after proper changes in major revision for publication in Materials. Some of my specific comments are below:
- Describe the novelty of the article made by the author? From the results of my evaluation, it seems that many similar published works adequately explain what you have raised in the current manuscript related to Mechanical properties of Titanium Alloy from Wire Arc Additive Manufacturing. If there are something others really new in this manuscript, please highlight it more clearly in the introduction section (line 33-113).
- The state of the art and the significance of the current study are not clearly present, the authors should highlight it more advanced in the introduction section (line 33-113).
- Since this manuscript evaluate metalic materials and have discussed regarding its biocompability (line 34-36), I would encourage and advise the authors to adopt some of the specific additional references related to metallic materials in medical implant application published by MDPI in the introduction section (line 33-113) as follow:
-
- Tresca Stress Simulation of Metal-on-Metal Total Hip Arthroplasty during Normal Walking Activity. Materials (Basel). 2021, 14, 7554. https://doi.org/10.3390/ma14247554
- The Effect of Bottom Profile Dimples on the Femoral Head on Wear in Metal-on-Metal Total Hip Arthroplasty. Journal of Functional Biomaterials. 2021, 12, 38. https://doi.org/10.3390/jfb12020038
- In the materials and methods section (line 114-182), the authors should add one systematic figure to illustrate the workflow of experimental testing in the present study to make the reader more interested and easier to understand rather than only using dominant text to explain.
- It is crucial to explain more clearly regarding the reason of dimension used for titanium materials based specimen in present experimental testing.
- The author must provide a detailed specification and use condition more detail regarding all tools used in the research carried out so that the reader can estimate the accuracy and differences in the results that the authors describe due to the use of different tools in future studies.
- The authors are advised to compare the results they obtain with previous similar/identical studies if it is possible.
- In the last paragraph before conclusion section (after line 449), the authors should add of one paragraph about the limitations of the presented article.
- The conclusion (line 450-496) of the present manuscript is too long and not solid. Further elaboration is needed. Also, make it intho paragraph, not point-by-point as in present form.
- Further research needs to be explained in the conclusion section (line 450-496).
- In the whole of the manuscript, the authors sometimes made a paragraph only consisting of one or two sentences that made the explanation not clearly understood. The authors need to extend their explanation to become a more comprehensive paragraph. In one paragraph, it is recommended to consist of at least 3 sentences with 1 sentence as the main sentence and the other sentences as supporting sentences.
- I see some errors on English in some areas of the present manuscript. To improve the quality of English used in this manuscript and make sure English language, grammar, punctuation, spelling, and overall style are correct, further proofreading is needed. As an alternative, the authors can use the MDPI English proofreading service for this issue.
- Please make sure the authors have used the Materials, MDPI format correctly. The authors can download published manuscripts by Materials, MDPI, and compare them with the present author's manuscript to ensure typesetting is appropriate. For example: Authors Contribution, Funding, Institutional Review Board Statement, Informed Consent Statement, Data Availability Statement, Acknowledgments, and Conflicts of Interest Information is not provided.
I am pleased to have been able to review the author's present manuscript. Hopefully, the author can revise the current manuscript as well as possible so that it becomes even better. Good luck for the author's work and effort.
Best regards,
The Reviewer
Round 2
Reviewer 1 Report
The revised version of the article is much better. In general, the authors corrected all my comments.
Reviewer 2 Report
Dear Authors,
Thank you for responding well to all review comments.
Reviewer 3 Report
Dear Tian et al.,
The revised manuscript is improved and well written by the authors. One thing I need to be mentioned, in line 117-118 the authors stated “This model provides a reference 117 for the numerical simulation of TC11 titanium alloy…”. The numerical simulation also can be solved using a computational model based on finite element analysis that is widely applied today. To enhance the explanation please report this aspect in the introduction and/or discussion section quoting reference published by MDPI as follow:
Computational Contact Pressure Prediction of CoCrMo, SS 316L and Ti6Al4V Femoral Head against UHMWPE Acetabular Cup under Gait Cycle. J. Funct. Biomater. 2022, 13, 64. https://doi.org/10.3390/jfb13020064
Best regards,
The Reviewer